

# Improvement of growth and bacoside production in *Bacopa monnieri* through induced autotetraploidy with colchicine

Phithak Inthima[1,2,*] and  Kawee Sujipuli[2,3,*]

[1] Plant Tissue Culture Research Unit, Department of Biology, Faculty of Science, Naresuan University, Phitsanulok, Thailand
[2] Center of Agricultural Biotechnology, Naresuan University, Phitsanulok, Thailand
[3] Department of Agricultural Science, Faculty of Agriculture Natural Resources and Environment, Naresuan University, Phitsanulok, Thailand
[*] These authors contributed equally to this work.

## ABSTRACT

*Bacopa monnieri* is a medicinal herb that is increasing in demand in Thailand. However, the lack of high-bacoside cultivars has limited pharmaceutical utilization and production. Here, chromosome doubling in *B. monnieri* was attempt to improve biomass and bacoside content in its seedling. Nodal segments were treated with colchicine (0, 0.025, 0.05, 0.075, 0.1, and 0.5% w/v) for 24 or 48 h before transferring to multiple shoot induction medium (1/2 MS medium supplemented with 0.2 mg $L^{-1}$ BAP). Of 326 tested clones, 18 and 84 were mixoploids and autotetraploids, respectively. The highest autotetraploid-induction percentage (14.6%) was found after treated with 0.5% (w/v) colchicine, and 48 hours exposure. From 28 selected autotetraploid clones, 21 and 13 have significantly higher fresh and dry weight compared to the diploid clone, respectively. The maximum fresh and dry weight of autotetraploid plants was 2.8 and 2.0-time higher than diploid plants, respectively. Moreover, the maximum total bacoside content (1.55 mg $plant^{-1}$) was obtained from an autotetraploid plant, which was 2.3-fold higher than the level in diploid plants. These novel autotetraploids have the potential to be developed as resources for value-added improvements in the medicinal and pharmaceutical industries.

## INTRODUCTION

*Bacopa monnieri* (commonly known as waterhyssop or brahmi in Ayurvedic medicine) is a small succulent herb that grows in marshy and moist tropical regions (*Tripathi et al., 2012*). It has been extensively utilized as a traditional Ayurvedic medicine to enhance memory and brain rejuvenation, promote longevity and improve intellectual functions (*Gubbannavar et al., 2012*). Various pharmacological effects of brahmi have been reported including anti-parkinson's (*Jansen et al., 2014*), antidepressant (*Rauf et al., 2014*), anti-anxiety (*Pandareesh et al., 2014*), and as an Alzheimer's disease treatment (*Uabundit et al., 2010*). The bioactive compounds of brahmi are mainly associated with triterpenoid saponins as complex mixtures (*Deepak et al., 2005*). Of these, bacoside A

Corresponding authors
Phithak Inthima, phithaki@nu.ac.th
Kawee Sujipuli, kawees@nu.ac.th

is the core active compound which acts as a memory booster (*Ramasamy et al., 2015*). Biosynthetic pathway of triterpenoid saponins, like sterols, had been hypothesized through both the mevalonic acid (MVA) and the methylerythritol phosphate (MEP) pathways (*Augustin et al., 2011*), resulting in the formation of intermediate substance as the farnesyl pyrophosphate (*FPP*) by two enzyme actives *of* dimethylallyl diphosphate (*DMAPP*), *and* geranyl pyrophosphate (*GPP*) (*Acharya et al., 2009*). This substance converts into a $C_{30}$ molecule squalene by squalene synthase (*SQS*), and its molecule subsequently cyclizes into triterpene skeletons by 2, 3-oxidosqualene (*OSCs*) (*Niu et al., 2014*). Finally, their skeletons are catalyzed by cytochrome P450-dependent monooxygenases (*CYP450*) and glycosyltransferase (*GT*) to produce triterpenoid saponins (*Abbassi et al., 2015*). Since the triterpenoid saponin pathway involved with many genes, so the improvement of a new variety for increasing saponin by pollination breeding is time consuming and extremely difficult (*Samaddar et al., 2012*).

Many products containing brahmi are widely available today. To satisfy the high commercial demand, 1,000 tons of raw plant material were used by herbal industries in 2011 (*Muthiah et al., 2013*), with most of this collected from the natural resources (*Tewari, 2000*) and this led to the rapid depletion of wild natural plants. However, cultivation of brahmi is restricted to certain agroclimatic conditions in wetlands and muddy shores, and natural regeneration through seeds or cuttings is slow and requires good irrigation facilities. Moreover, improvement of new varieties of brahmi by conventional breeding is time consuming and difficult since the plant has a small flower and a high number of chromosomes, $2n = 2x = 64$ (*Samaddar et al., 2012*). Therefore, new techniques have been developed to improve brahmi production. One of the most popular techniques is elicitation with elicitors (physical agent, biological or chemical substances) treatments (*Sharma et al., 2013*) to increase bacoside content. However, since there are few reports concerning biosynthetic pathways in brahmi, the effects and functions of elicitation are difficult to predict. One technique to solve this limitation involves creating new sustainable varieties of brahmi through polyploidization using colchicine treatment because it is a rapid, cost-effective, efficient and safe method (*Sattler et al., 2016*). Moreover, previous studies have reported that tetraploid plants exhibit increased heterosis (hybrid vigor) such as increased oil content in *Ocimum basilicum* (*Omidbaigi et al., 2010*), wider and thicker leaves in *Fagopyrum tataricum* (*Wang et al., 2017a*), and increased root size in *Echinacea purpurea* (*Chen et al., 2016*). Previous studies have shown that the brahmi polyploids have been improved by enlarging the flower size (*Escandón et al., 2006*) and the shoot (*Sangeetha & Ganesh, 2011*), this is possible that the biomass and bacoside yield could be improved by polyploidization. In this study, chromosome doubling in *B. monnieri* was attempted to improve biomass and bacoside content in its seedling. The results may be beneficial for further genetic improvements to increase the medicinal value of brahmi and raw drug production.

## MATERIALS & METHODS

### Plant material and colchicine treatment

Nodal segments (one cm in length) of diploid *B. monnieri* were cut from one month in vitro culture plantlets. These segments were treated with 1/2 MS (*Murashige & Skoog, 1962*) liquid medium supplemented with one of six different colchicine concentrations (0.025, 0.05, 0.075, 0.1 or 0.5% w/v) in parallel with nodal segments in 1/2 MS only as the control. All samples were incubated with shaking at 100 rpm for 24 or 48 h under the condition of $25 \pm 2\,°C$ in a 10-hour photoperiod with the light intensity of $20\ \mu mol\ m^{-2}\ s^{-1}$ provided by warm white LED lamps. After treatment, all colchicine treated explants were rinsed with sterilized distilled water five times and then transferred to solidified 1/2 MS medium supplemented with $0.2\ mg\ L^{-1}$ 6-Benzylaminopurine (BAP, *Sharma et al., 2010*). The cultures were kept for 8 weeks under the same standard condition as mentioned above, but without shaking.

### In vitro regenerant screening and flow cytometry analysis

Shoots of eight-week-old treated regenerants were excised and transferred to fresh solidified 1/2 MS medium without hormone, and cultured under the same standard conditions for 6 weeks. The putative polyploid plants were screened using the criteria of bigger and thicker leaves and stems, as compared to diploid plantlets (*Escandón et al., 2006*). These plants were further confirmed for polyploidy level by a flow cytometry assay. A leaf sample (100 mg) was chopped with a razor blade in a petri dish containing one mL nuclei extraction buffer (200 mM Tris, 4 mM $MgCl_2.6H_2O$, 0.5% (v/v) Triton X-100, pH 7.5) according to the method of *Pfosser et al. (1995)*. The nuclei suspension was filtered through a $40\ \mu m$ nylon net (Merck Millipore Ltd., Germany) and stained using the Muse[TM] Cell Cycle Kit (Merck KGaA, Germany). The polyploidy level was analyzed on a Guava® easyCyte Flow Cytometer with InCyte[TM] software version 2.7 (Merck KGaA, Germany).

### Morphological observations in autotetraploid plants

The autotetraploid clones were in vitro proliferated on 1/2 MS for 8 weeks and the plantlets with the same height were transplanted into pots containing 1/2 Hoagland solution (HS, (*Hoagland & Arnon, 1950*) and kept under greenhouse condition for 30 days. At least five plants per clone were grown and observed for biomass, height, leaf thickness, stem diameter and additionally quantifies for bacoside content. For measurement of leaf thickness and stem diameter, the 5th leaf and internode were cross section with a razor blade. The samples were observed and measured under light microscope. The measurements were undertaken with 5 measuring points per leaf or stem of five individual plants.

### Quantification of bacoside content

Bacoside content was quantified using the modified method of *Bansal et al. (2016)*. Briefly, aerial parts of 30-day-old hydroponically grown plants were dried at 40 °C for 2 days and then ground to a fine powder. The powder (0.1 g) was then extracted with three mL methanol by incubated at room temperature for 1 h, sonicated for 15 min and then incubated in the dark at 4 °C for 5 min. After centrifugation at 8,000 rpm for 5 min,
the supernatant was filtered through a 0.45-µm nylon filter (Tianjin Fuji Science and Technology Co., Ltd., China). Bacoside content in the filtered extracts was analyzed on a HPLC system (Shimadzu, Japan) equipped with Purospher®STAR-RP-18 endcapped (5 µm) LiChroCART®250-4.6 column (Merck, Germany). The chromatographic condition was as follows: (i) the mobile phase was 0.2% (v/v) aqueous phosphoric acid, pH 3.0 and acetonitrile (65:35 v/v), (ii) the flow rate was one mL min$^{-1}$, and (iii) bacoside was detected at 205 nm using a SPD-10A VP UV-Vis detector (Shimadzu, Japan). The bacoside content in the plant extracts was calculated by a linearity equation of peak areas against known concentrations bacosides (Sigma-Aldrich, US).

## Statistical analysis

The experiment was conducted using a complete randomized design. The differences in fresh weight, dry weight and height of autotetraploid and diploid plants were compared by $t$-test. Other data were analyzed using one-way analysis of variance (ANOVA) and Duncan's multiple range test (DMRT).

## RESULTS AND DISCUSSION

### Effect of colchicine on growth and phenotypic variation of regenerated brahmi

Eight weeks after transfer of the colchicine-treated nodal segments to new 1/2 MS, the highest percentage of shoot multiplication (22.0 regenerants per explant) was found in the segments treated with 0.025% colchicine for 24 h (Table 1). These shoots were found to be healthier than the control and other treatments (Fig. 1A). Shoot multiplication and shoot length significantly decreased with increasing concentrations and exposure times of colchicine as compared to untreated explants (Table 1, Fig. 1A). Increasing concentrations and exposure times of colchicine have been shown to inhibit growth and shoot regeneration of explants (*Javadian et al., 2017*; *Wang et al., 2017a*). Moreover, higher concentrations of colchicine with longer exposure times can be toxic to plants leading to tissue abnormalities (*Megbo, 2010*) and dead (*Mondin et al., 2018*).

Some regenerants of colchicine-treated explants showed distinct morphological feature changes in leaf and stem such as albinoism (Fig. 1B), a spiral stem (Fig. 1C), variegated leaves (Fig. 1D), whorls of three (Fig. 1E) and four (Fig. 1F) leaves, and a fasciated stem (Fig. 1G). These results indicated that colchicine treatment induced phenotypic changes in the regenerated brahmi. Several published studies have reported that colchicine-induced wide morphological variations such as leaf arrangement and size in *Platycodon grandiflorus* (*Wu et al., 2011*) and *Hebe* (*Gallone et al., 2014*), stomatal size in *Pogostemon cablin* (*Yan et al., 2016*) and cassava (*Mondin et al., 2018*), and leaf, flower, and seed size in *Fagopyrum tataricum* (*Wang et al., 2017a*).

### Flow cytometry analysis for determining polyploidy level in colchicine-treated plantlets

From a total of 1,262 regenerants, 326 putative autotetraploid clones were selected based on criteria of bigger and thicker leaves and stems as compared to diploid plantlets. Polyploidy

**Table 1 Effect of concentration and treatment duration of colchicine on polyploidy induction in nodal segments of *Bacopa monnieri*.**

| Time of treatment (h) | Colchicine (% w/v) | No. of regenerant | | Flow cytometry analysis | | | | |
|---|---|---|---|---|---|---|---|---|
| | | | | No. of tested regenerants | Mixoploid | | Tetraploid | |
| | | Total | Average[a] | | Number | %[b] | Number | %[b] |
| | 0 | 97 | 12.1 ± 3.4 b | 27 | 0 | 0 | 0 | 0 |
| | 0.025 | 176 | 22.0 ± 9.4 a | 25 | 1 | 0.6 | 0 | 0 |
| | 0.05 | 112 | 14.0 ± 4.4 ab | 29 | 1 | 0.9 | 3 | 2.7 |
| 24 | 0.075 | 149 | 18.6 ± 5.8 ab | 34 | 2 | 1.3 | 14 | 9.4 |
| | 0.1 | 120 | 15.0 ± 4.6 ab | 35 | 5 | 4.2 | 9 | 7.5 |
| | 0.5 | 115 | 14.4 ± 8.8 ab | 33 | 1 | 0.9 | 7 | 6.1 |
| | Total[c] | 672 | | 156 | 10 | 1.4 | 33 | 4.9 |
| | 0 | 81 | 10.1 ± 5.6 b | 26 | 0 | 0 | 0 | 0 |
| | 0.025 | 151 | 18.9 ± 2.9 ab | 34 | 1 | 0.7 | 2 | 1.3 |
| | 0.05 | 143 | 17.9 ± 10.5 ab | 35 | 0 | 0 | 15 | 10.5 |
| 48 | 0.075 | 108 | 13.5 ± 5.0 ab | 33 | 0 | 0 | 11 | 10.2 |
| | 0.1 | 99 | 12.4 ± 2.9 b | 36 | 5 | 5.1 | 10 | 10.1 |
| | 0.5 | 89 | 11.1 ± 4.5 b | 32 | 2 | 2.2 | 13 | 14.6 |
| | Total[c] | 590 | | 170 | 8 | 1.4 | 51 | 8.6 |
| **Total[c]** | | **1,262** | | **326** | **18** | **1.4** | **84** | **6.7** |

Notes.
[a] Average value represents as mean ± SE of a number of regenerants per nodal segment, calculated from eight nodal segments at 8 weeks after culture. Different letters within the same column indicate significant differences, analyzed by Duncan's Multiple Range Test (DMRT) at $p \leq 0.01$.
[b] Percentage value was calculated from number of mixoploid or tetraploid per total number of regenerants in each treatment.
[c] Data were summarized by all regenerants from colchicine treatment combinations.

level was analyzed by flow cytometry. The DNA content peak of a diploid control plant ($2x$; 2n = 64) was set at channel 100 (Fig. 2A) and the peak of an autotetraploid plant ($4x$; 2n=128) was measured at channel 200, which was twofold intensity compared to that of diploid plantlets (Fig. 2B). Of 326 clones tested, only 18 and 84 clones were mixoploid (a plant composed of both $2x$ and $4x$ cell, Fig. 2C) and autotetraploid, respectively (Table 1). Among the twelve colchicine treatment combinations, the mixoploid- and autotetraploid-induction efficiencies (%) ranged between 0.0–5.1% and 0-14.6%, respectively (Table 1). The highest autotetraploid-induction efficiency (14.6%) was found in the combination of treated with 0.5% (w/v) colchicine and exposure for 48 h (Table 1). Moreover, the highest mixoploid induction (5.1%) was found after treatment with 0.1% (w/v) colchicine and exposure of 48 h (Table 1). In contrast, treatment of explants with 0.025% colchicine for 24 h did not induce autotetraploid in any regenerated plantlets (Table 1). In this study, nodal segments of brahmi treated with 0.5% colchicine solution for 48 h was the most useful condition to generate the highest number of autotetraploid plantlets. Because colchicine plays an important role in inhibiting microfilament and microtubule formation and disturbs mitosis at the anaphase stage, chromosomes do not migrate to the opposite ends of the cell during cell division, resulting in chromosome doubling in the cell (*Pour et al., 2012*). Furthermore, a recent study reported that colchicine affected

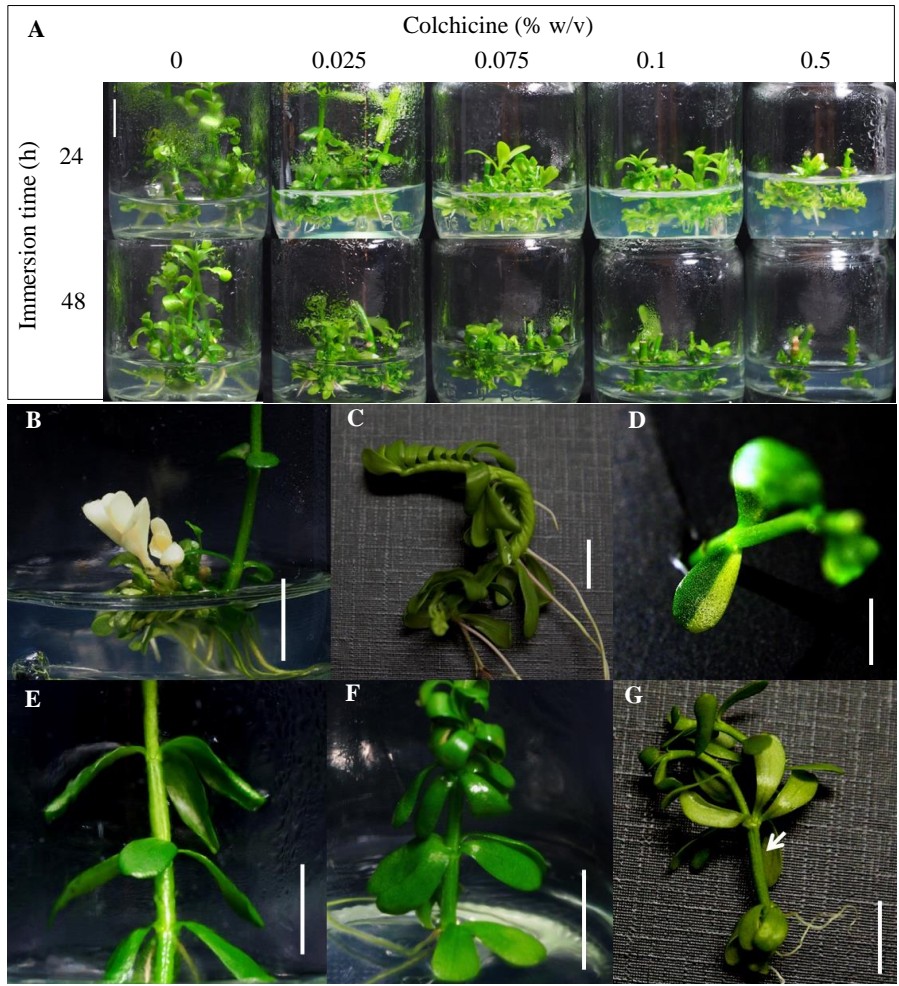

**Figure 1** **Effect of colchicine on shoot multiplication and phenotypic variation of regenerated *Bacopa monnieri*.** After colchicine-treated nodal segments of *B. monnieri*, shoot multiplications of regenerats were observed after cultured on 1/2 MS for 8 weeks (A). Some morphological variations were observed in regenerated *B. monnieri* after colchicine treatment such as albino leaf and stems (B), spiral stem (C), variegated leaves (D), whorl of three leaves (E), whorl of four leaves (F), and a fasciated stem (arrow) with two apical shoots (G). Bars indicates 1 cm.

numerous genes related to the spindle, the chromosomal kinetochore, vesicles, cellulose and the processes of cytoplasm movement, chromatid segregation, as well as membrane and cell wall development (*Zhou et al., 2017*). For this reason, colchicine is a highly potent antimitotic agent which has been successfully used for inducing polyploidy in many plant species (*Pour et al., 2012*; *Wang et al., 2017a*; *Mondin et al., 2018*).

## Agricultural characters in autotetraploid plantlets

Of the 84 autotetraploids, 28 clones were selected (based on in vitro growth performances) and transplanted into 1/2 HS under greenhouse conditions for 30 days. The results showed that 21 and 13 examined autotetraploids clones had significantly higher fresh and dry weight compared to diploid clone, respectively (Figs. 3A and 3B). The highest fresh (1.50 g

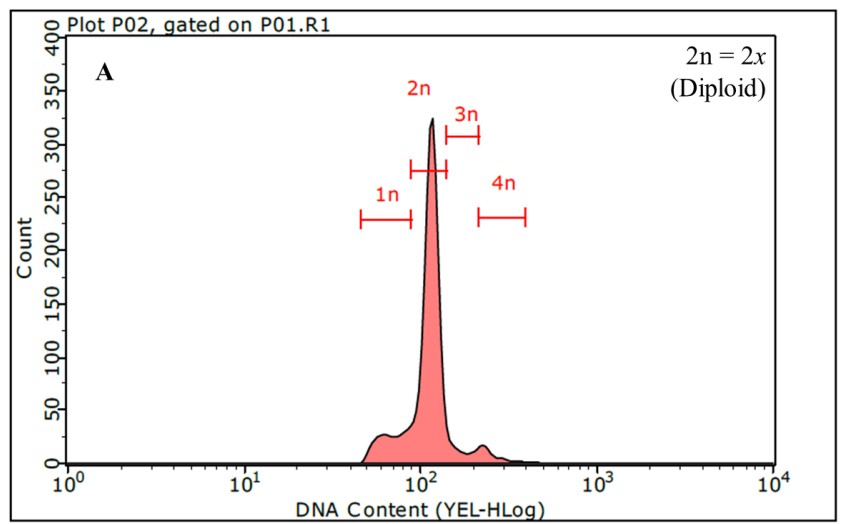

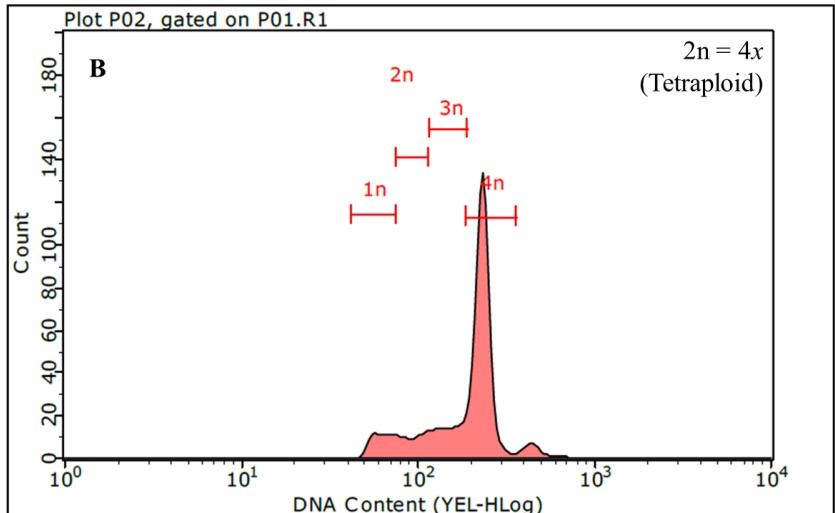

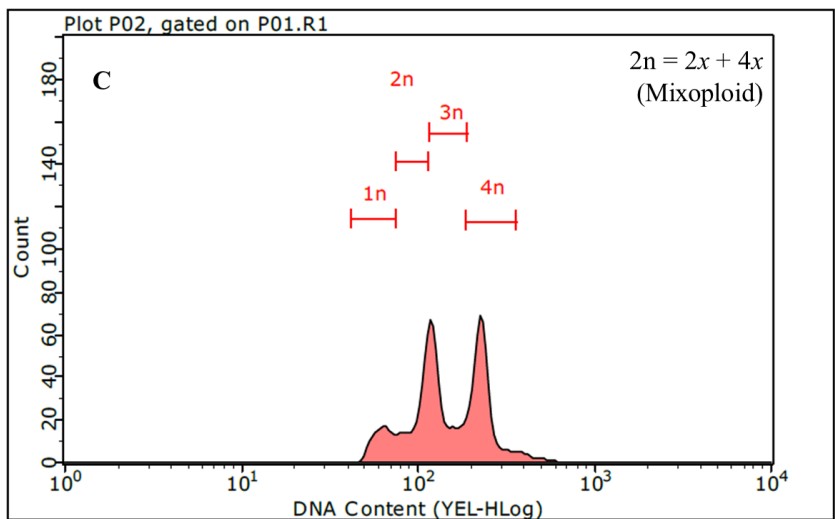

**Figure 2** **Identification of polyploid *Bacopa monnieri* using flow cytometry.** Histograms of diploid (A), autotetraploid (B) and mixoploid (C) are shown.

plant$^{-1}$) and dry (0.072 g plant$^{-1}$) weight was found in autotetraploid-plant clone 4$x$-103 (Fig. 4A), which was around 2.8 and 2.0-time higher than diploid plants, respectively (Figs. 3A and 3B). This was probably due to the autotetraploid having thicker leaves (Figs. 4B and 4D) and larger stem diameter (Figs. 4C and 4E) than its diploid progenitor. In contrast, shoot height in six autotetraploid clones slightly increased (16.0 to 18.8 cm) more than diploid plant (14.2 cm), and almost half (16 of 28) of the tetraploid clones were not significantly different from the diploid plant controls (Fig. 3C).

The results clearly showed that most of the selected autotetraploid plants had significantly greater fresh weight, dry weight, and shoot height than diploid plants. The reasons of this were possible that colchicine-side and double-gene-dosage effects, that might either increase or decrease biomass and metabolite production of polyploids (Javadian et al., 2017; Salma et al., 2018). In some cases, polyploids were much more effective in superior vegetative-tissue shapes and sizes than diploids, which may be the reason for an increase in biomass yields (Wang et al., 2017a). Another reason for this was that colchicine might have induced morphological variations such as leaf and stem thickness, causing enhanced fresh weight and dry weight (Niu et al., 2016). Moreover, chromosome doubling might have induced the differential gene expression between diploid and synthesized tetraploid, therefore affecting the various phenotypic traits (Liqin et al., 2019). The previous study has shown differences in gene expression between diploid and triploid of *Populus*, involved in carbohydrate and lipid metabolism in triploid, reflected a more significant increase in carbon metabolism and utilization efficiency than diploids, which may be responsible for the fast-growing trait observed in triploids (Cheng et al., 2015). The organ sizes (plant height, leaf area, and petiole length) were significantly superior in polyploids than diploids *Populus* (Liqin et al., 2019), since polyploids might be up-regulated the growth-regulating factor (GRF) gene family, played vital roles in the control of plant growth development (Wu et al., 2014).

## Accumulation of bacoside content in autotetraploid explant

Of the 28 autotetraploid clones, 14 representatives were selected using the criteria of higher fresh weight and/or dry weight than diploid plants. Total bacoside content of 14 representatives was quantified using HPLC as shown in Table 2. Only one autotetraploid clone (4$x$-59, Fig. 4C) had a significantly ($p \leq 0.05$) higher total bacoside contents than the diploid clones. However, two autotetraploid clones (4$x$-59 and 4$x$-103) had maximum amounts of bacoside at 1.55 and 1.10 mg plant$^{-1}$ respectively, as compared to diploid clone (0.68 mg plant$^{-1}$) and significantly ($p \leq 0.05$) higher than diploid clone by 2.3 and 1.6 folds, respectively. This suggested that their polyploidy may be associated with improving secondary metabolite content in brahmi.

Since the biosynthetic pathways of secondary metabolites involve many steps and require several enzymes, it was hypothesized that doubling chromosome number in autopolyploid plants may increase the number of gene copies. This may lead to increased enzymatic synthesis and metabolic activity, which may provide additional substrates for enhancing secondary plant metabolites (Salma et al., 2017). Previous publications have reported more abundant proteins related to the biosynthesis of secondary metabolites

Inthima and Sujipuli (2019), *PeerJ*, DOI 10.7717/peerj.7966

**Table 2 Accumulation of bacoside contents in autotetraploid plants of *Bacopa monnieri* after growth in 1/2 HS for 30 days, analyzed using HPLC technique.**

| Clone no. | Content (%DW) | | | | | Yield (mg plant $^{-1}$)[a] | | | | |
|---|---|---|---|---|---|---|---|---|---|---|
| | Bacoside A | Bacopaside II | Bacopaside X | Bacopasaponin C | Total bacoside | Bacoside A | Bacopaside II | Bacopaside X | Bacopasaponin C | Total bacoside |
| 2x | 0.22 ± 0.03[ab] | 0.66 ± [a–c] | 0.17 ± 0.05[b] | 0.72 ± 0.12[bc] | 1.77 ± 0.31[ab] | 0.08 ± 0.01[b–d] | 0.25 ± 0.05[c–e] | 0.06 ± 0.02[bc] | 0.27 ± 0.05[b–d] | 0.68 ± 0.13[b–d] |
| 4x-9 | 0.12 ± 0.02[c–g] | 0.54 ± 0.12[bc] | 0.57 ± 0.11[a] | 0.25 ± 0.06[d–f] | 1.48 ± 0.31[b] | 0.05 ± 0.01[c–f] | 0.23 ± 0.05[de] | 0.25 ± 0.04[a] | 0.11 ± 0.02[c] | 0.64 ± 0.13[b–d] |
| 4x-19 | 0.14 ± 0.03[b–g] | 0.67 ± 0.13[a–c] | 0.00 ± 0.00[b] | 0.62 ± 0.13[b–e] | 1.43 ± 0.29[b] | 0.06 ± 0.01[b–f] | 0.30 ± 0.06[c–e] | 0.00 ± 0.00[c] | 0.28 ± 0.06[b–d] | 0.64 ± 0.13[b–d] |
| 4x-20 | 0.05 ± 0.01[g] | 0.33 ± 0.05[c] | 0.37 ± 0.07[a] | 0.27 ± 0.06[d–f] | 1.01 ± 0.19[b] | 0.02 ± 0.00[f] | 0.14 ± 0.02[e] | 0.16 ± 0.02[ab] | 0.11 ± 0.02[d] | 0.43 ± 0.07[d] |
| 4x-23 | 0.20 ± 0.02[a–c] | 0.85 ± 0.14[ab] | 0.00 ± 0.00[b] | 0.84 ± 0.12[b] | 1.89 ± 0.28[ab] | 0.11 ± 0.01[ab] | 0.46 ± 0.05[a–c] | 0.00 ± 0.00[c] | 0.47 ± 0.03[b] | 1.04 ± 0.08[bc] |
| 4x-40 | 0.07 ± 0.01[e–g] | 0.34 ± 0.04[c] | 0.38 ± 0.05[a] | 0.15 ± 0.03[f] | 0.94 ± 0.12[b] | 0.05 ± 0.01[c–f] | 0.23 ± 0.05[de] | 0.26 ± 0.05[a] | 0.11 ± 0.03[d] | 0.64 ± 0.14[b–d] |
| 4x-56 | 0.10 ± 0.02[d–g] | 0.48 ± 0.10[bc] | 0.54 ± 0.11[a] | 0.26 ± 0.05[d–f] | 1.37 ± 0.29[b] | 0.04 ± 0.01[d–f] | 0.16 ± 0.04[e] | 0.19 ± 0.04[a] | 0.09 ± 0.02[d] | 0.47 ± 0.11[d] |
| 4x-59 | 0.25 ± 0.05[a] | 1.05 ± 0.24[a] | 0.12 ± 0.01[b] | 1.30 ± 0.29[a] | 2.73 ± 0.60[a] | 0.15 ± 0.03[a] | 0.60 ± 0.13[a] | 0.07 ± 0.01[bc] | 0.74 ± 0.15[a] | 1.55 ± 0.31[a] |
| 4x-62 | 0.17 ± 0.04[a–d] | 0.75 ± 0.15[a–c] | 0.00 ± 0.00[b] | 0.68 ± 0.16[b–d] | 1.59 ± 0.34[b] | 0.09 ± 0.02[bc] | 0.40 ± 0.07[a–d] | 0.00 ± 0.00[c] | 0.36 ± 0.07[bc] | 0.84 ± 0.16[b–d] |
| 4x-88 | 0.09 ± 0.01[d–g] | 0.40 ± 0.02[bc] | 0.39 ± 0.02[a] | 0.19 ± 0.02[ef] | 1.07 ± 0.06[b] | 0.05 ± 0.00[c–f] | 0.19 ± 0.01[de] | 0.19 ± 0.01[a] | 0.09 ± 0.00[d] | 0.53 ± 0.02[cd] |
| 4x-94 | 0.15 ± 0.04[b–e] | 0.77 ± 0.19[a–c] | 0.00 ± 0.00[b] | 0.81 ± 0.22[b] | 1.74 ± 0.44[ab] | 0.07 ± 0.01[b–e] | 0.35 ± 0.08[b–e] | 0.00 ± 0.00[c] | 0.37 ± 0.09[bc] | 0.80 ± 0.19[b–d] |
| 4x-98 | 0.06 ± 0.02[e–g] | 0.42 ± 0.12[bc] | 0.40 ± 0.12[a] | 0.20 ± 0.06[ef] | 1.08 ± 0.32[b] | 0.04 ± 0.01[d–f] | 0.24 ± 0.07[de] | 0.23 ± 0.07[a] | 0.11 ± 0.04[d] | 0.62 ± 0.20[b–d] |
| 4x-99 | 0.15 ± 0.03[b–f] | 0.70 ± 0.15[a–c] | 0.00 ± 0.00[b] | 0.38 ± 0.09[c–f] | 1.23 ± 0.27[b] | 0.06 ± 0.02[c–f] | 0.31 ± 0.08[c–e] | 0.00 ± 0.00[c] | 0.17 ± 0.05[cd] | 0.54 ± 0.14[cd] |
| 4x-103 | 0.13 ± 0.03[b–g] | 0.68 ± 0.14[a–c] | 0.00 ± 0.00[b] | 0.61 ± 0.14[b–e] | 1.42 ± 0.31[b] | 0.10 ± 0.02[b] | 0.53 ± 0.10[ab] | 0.00 ± 0.00[c] | 0.47 ± 0.10[b] | 1.10 ± 0.22[ab] |
| 4x-115 | 0.06 ± 0.03[fg] | 0.42 ± 0.10[bc] | 0.53 ± 0.14[a] | 0.38 ± 0.10 c-f | 1.39 ± 0.36[b] | 0.03 ± 0.01[ef] | 0.20 ± 0.04[de] | 0.26 ± 0.06[a] | 0.19 ± 0.04[cd] | 0.68 ± 0.16[b–d] |

**Notes.**

The data represent the mean ± SE of three biological replicates. Different letters within the same column indicate significant differences analyzed by DMRT at $p \leq 0.05$.

[a]The yield of bacoside was calculated on the basis of dry weight of individual plant.

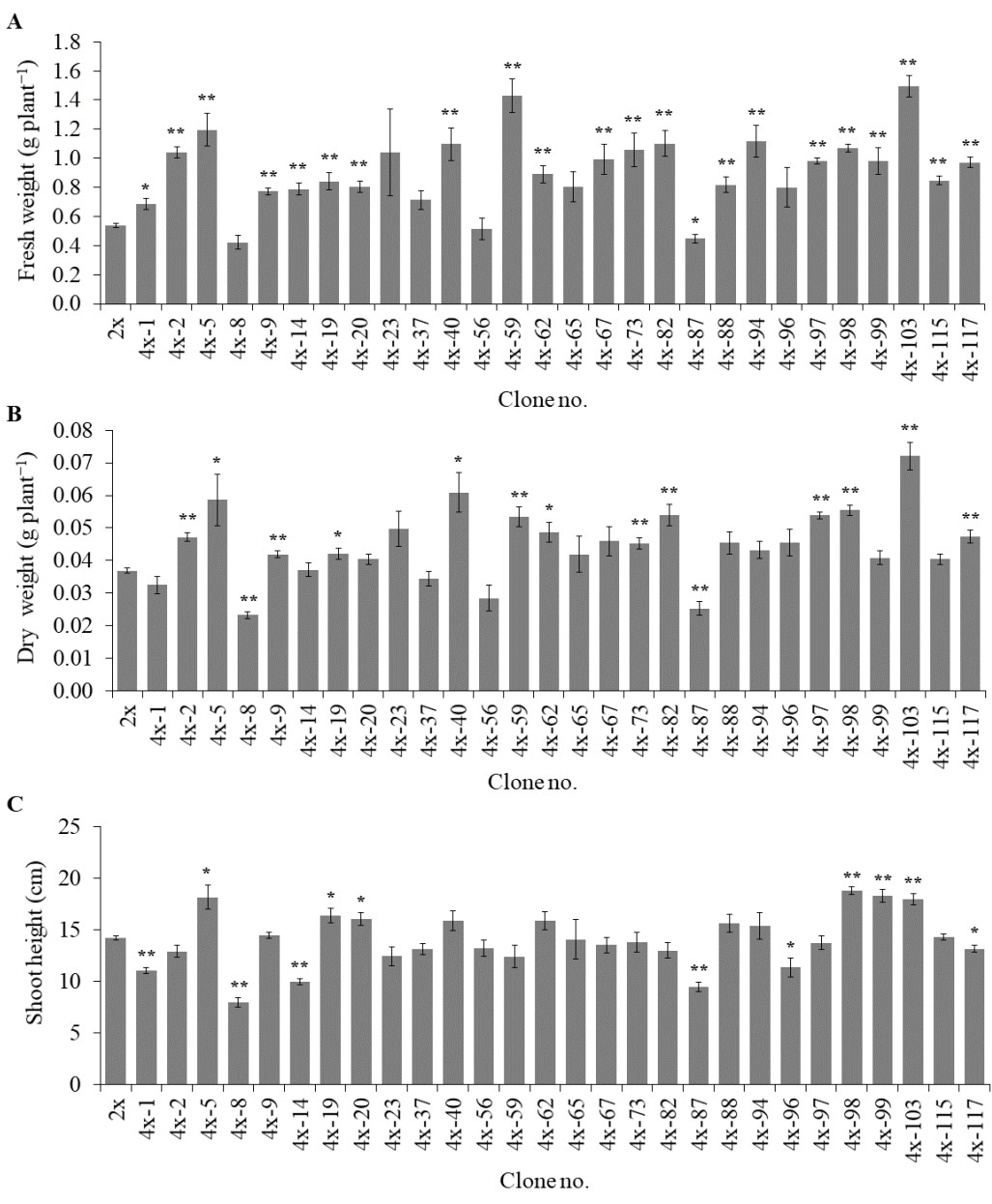

**Figure 3** **Fresh weight (A), dry weight (B) and shoot height (C) in selected autotetraploid plants of *Bacopa monnieri* after growth in 1/2 HS for 30 days.** Data are the mean of five plants and the error bar represents SE. The single asterisk (*), and double asterisk (**) indicate significant difference ($p \leq 0.05$) and highly significant difference ($p \leq 0.01$) of autotetraploid plants ($4x$) compared to diploid plant ($2x$) by *t*-test, respectively.

in autotetraploid than diploid *Paulownia australis* (*Wang et al., 2017b*). Similarly, the polyploidy-mediated change in phenotypic traits may appear as an effect of secondary metabolite biosynthesis. For example, enhancement of the leaf area and leaf thickness in autotetraploid *Stevia rebaudiana* were positively affected glycoside biosynthesis as a result of increased photosynthesis (*Hegde et al., 2015*). Similarly, polyploidization enhances

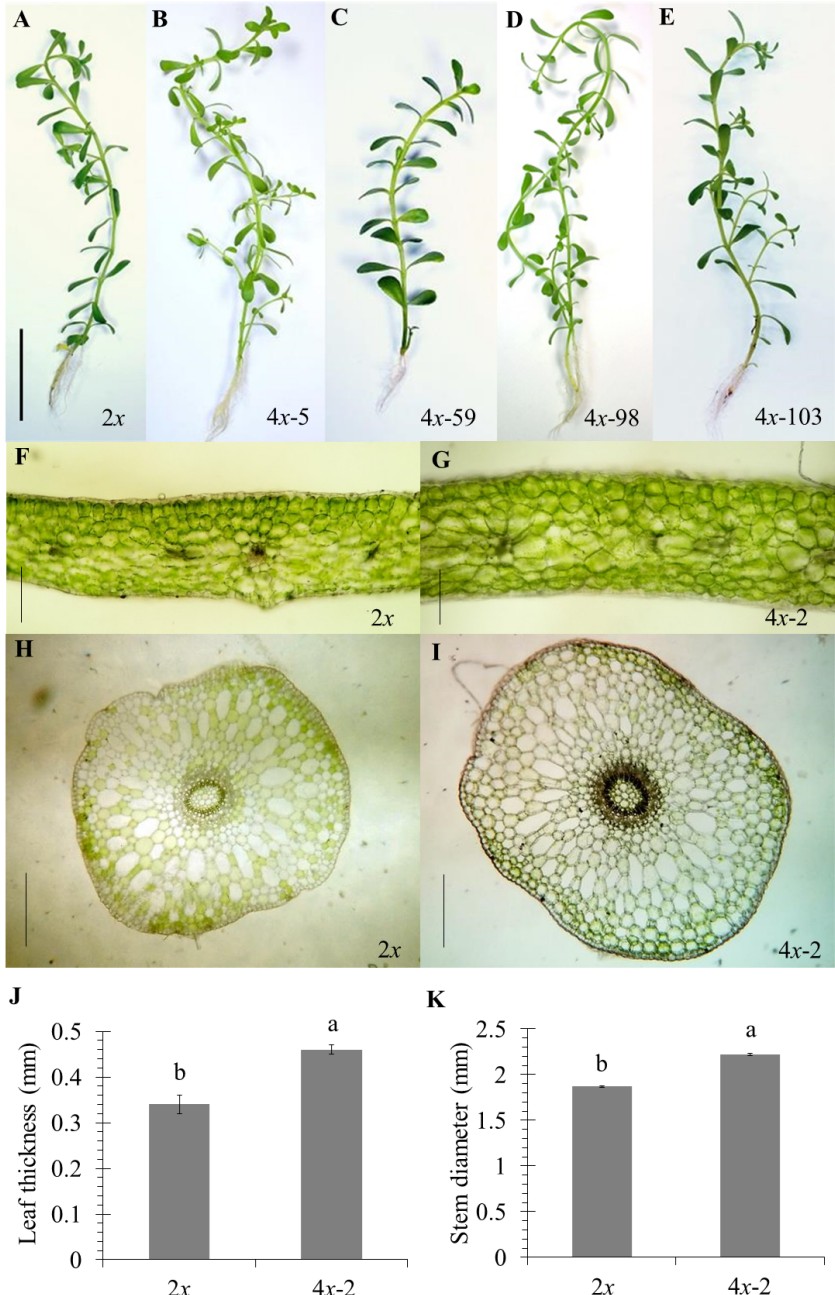

**Figure 4** **Morphological characters in diploid (A) and five representative autotetraploid (B–E) plants of *Bacopa monnieri* after growth in 1/2 HS for 30 days; bar indicates five cm.** Cross sections represent leaf thickness (F–G) and stem diameter (H–I) of diploid (F and H) and one representative sample of tetraploid (G and I) *B. monnieri*, bars indicate 250 μm. Measurement of leaf thickness (J) and stem diameter (K) were undertaken with 5 measuring points per leaf or stem of five individual plants after growth in 1/2 HS for 30 days. The different letters above the bars indicate significant difference at $p \leq 0.01$ according to DMRT.

the content of triterpenoid in *Jatropha curcas* (*Niu et al., 2016*), patchouli alcohol in *Pogostemon cablin* (*Yan et al., 2016*), and flavonoid in *Fagopyrum tataricum* (*Wang et al., 2017a*) as compared to diploid plants. Additionally, it has been reported that the level of artemisinin in tetraploids of *Artemisia annua* was higher than diploid due to a significant increase of artemisinin metabolite-specific genes (*Lin et al., 2011*). Likewise, podophyllotoxin production in tetraploid *Linum album* was increased over diploid due to higher expression of genes related to its biosynthesis (*Javadian et al., 2017*). So, the enhancement of secondary metabolite production of tetraploid compared to diploid might be that the expression of crucial biosynthesis genes in tetraploid was increased.

## CONCLUSIONS

This present study demonstrated that colchicine-mediated technique was successfully used in in vitro induction of autotetraploidy in *B. monnieri*. Most autotetraploid plants were superior in agricultural characteristics (such as fresh weight, dry weight and total bacoside content) than the original parental diploid plants. This suggests that the novel autopolyploids developed here may be of use in improving the quality and quantity of medicinal plant raw material, to the benefit of the medical and pharmaceutical industries.

## ACKNOWLEDGEMENTS

The authors are appreciative for the comments and English language editing of Prof. Duncan Richard Smith, Ph.D., Institute of Molecular Biosciences, Mahidol University, Thailand.

### Funding

This work was supported by Naresuan University, Thailand (grant numbers P2560C433 and R2559B037). The funders had no role in study design, data collection and analysis, decision to publish, or preparation of the manuscript.

### Grant Disclosures

The following grant information was disclosed by the authors:
Naresuan University, Thailand: P2560C433, R2559B037.

### Competing Interests

The authors declare there are no competing interests.

### Author Contributions

- Phithak Inthima conceived and designed the experiments, performed the experiments, analyzed the data, prepared figures and/or tables, authored or reviewed drafts of the paper, approved the final draft.
- Kawee Sujipuli conceived and designed the experiments, analyzed the data, contributed reagents/materials/analysis tools, authored or reviewed drafts of the paper, approved the final draft.

## Data Availability

The raw measurements are available in the Supplemental Files.

## Supplemental Information

Supplemental information for this article can be found online at http://dx.doi.org/10.7717/peerj.7966#supplemental-information.

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
