# Peer review of "Improvement of growth and bacoside production in Bacopa monnieri through induced autotetraploidy with colchicine"

_PeerJ, doi:10.7717/peerj.7966_

## Round 0.1 · original submission · Major Revisions

This is an interesting study about the improvement of a medicinal compound by polyploidization. However, the authors need to thoroughly address each one of BOTH reviewers' comments before their paper can be reconsidered for possible publication. The major problem I see is that the methodology efficiency of polyploidy induction was not correctly tested. As pointed out by reviewer #1, the criteria of bigger and thicker leaves and stems as compared to diploid plantlets can bias the induction rate estimate.

Reviewer 1 ·

Basic reporting

The manuscript is written in a clear and unambiguous, professional English.
Although the standard structure was not used, the presented structure improves paper understanding. Figures and tables are clear.
Authors provided enough introduction and background; nevertheless, the aim was not answered. According with authors “… this study aimed to use an efficient method for inducing tetraploid in brahmi, ...”, however the methodology efficiency was not correctly tested. Authors should present a total induction rate or an equivalent measure. The selecting of “putative autotetraploid clones (…) based on criteria of bigger and thicker leaves and stems as compared to diploid plantlets” biasing the induction rate, not having a correct quantification of the efficiency of the methodology. Authors should analyze a random set of plantlets including all the types of phenotypes to calculate a correct induction rate and consequently the methodology efficiency. Authors need to improve the data to answer to this specific aim.

Experimental design

The submission describes an original research with high importance for the filing the identified knowledge gap. Although the research question is clear exposed, the experimental design is not adjusted to the first part of the aim, as exposed above. Therefore, I must decline to handle the manuscript.

Validity of the findings

Statistical approach should be improved upon. The two factors (colchicine concentration and exposure time) need to be tested for interaction between them. I suggested use, for example, a GLMM to test this and according with the result decide the next step.
Authors made a series of erroneous assumptions based on the incorrect approach used to quantify the technic efficiency, and thus they cannot conclude that “Most autotetraploid plants were superior in agricultural characteristics (such as fresh weight, dry weight and total bacoside content) than the original parental diploid plants” once they did not include in the analyses the possible presence of autotetraploids with similar morphology than diploids plantlets.

Additional comments

Nonetheless, I encourage authors to improve the manuscript and submit the article to another journal.

Reviewer 2 ·

Basic reporting

The background provided is relevant to the research problems/issues in the study. Most of the literature references are up-to-date or recent publications related to the issues addressed in the study. However, more details information needed, especially about the bacoside and its biosynthetic pathway.

Tables and figures are clear in general and relevant to the content of the article, of sufficient resolution, and appropriately described and labelled.

There are some minor languages and typographical errors found throughout the manuscript. Please refer to the revised manuscript (highlighted in red colour) for further improvement.

Experimental design

The research justification and objectives are clearly stated and discussed with reference to the related literature.

The methods described are commonly used in polyploidization study with colchicine. However, I suggest the authors improve the description of in vitro regenerant screening (line 92-104) by including chromosome counting method besides flow cytometry in determining the polyploid plants. In addition, the description of the histological study (see Fig. 4b&c) should also be included (line 105-110).

Validity of the findings

The data are analyzed correctly and interpreted in accordance with the objectives of the research.

The HPLC data are good, but the sample size might be small, as only half of the identified polyploid analysed, i.e. 14 out of 28 polyploids. The authors might consider providing a clear justification for the limited sample size used in the HPLC analysis.

Overall, the discussion of the research findings could be strengthened by giving more facts or examples from other related studies. For instance, how do colchicine-induced phenotypic changes as observed and reported in the manuscript? Likewise for bacoside content in polyploid plants.

Additional comments

This is one of the important studies that highlighting the use of colchicine in the production of autotetraploids in a medicinal herb, Bacopa monnieri. The authors had successfully identified 28 autotetraploids out of 326 tested clones. Undoubtley, this involves great efforts and resources by the authors. With some moderate changes, as stated above, I believe this manuscript is relevant and may contribute to filling the knowledge gap of polyploidization with colchicine.

Annotated reviews are not available for download in order to protect the identity of reviewers who chose to remain anonymous.

---

## Round 0.2 · Minor Revisions

The authors have carefully addressed the reviewers' comments. There are a few lingering problems with use of language, which Reviewer #2 highlighted in the attached manuscript. Reviewer #1 suggests the inclusion of a citation and highlighted some typographical errors. After these recommended, minor revisions, the manuscript will be acceptable for publication.

Reviewer 1 ·

Basic reporting

The manuscript is in accordance with the journal requirements, with introduction and background sustained by the bibliography and clear figures and tables.
However, there are some minor typographical errors:
- in the expression "2x" and "4x", the "x" should be in italic. All manuscript included figures and legends should be reviewed and corrected;
- lines 219 and 222 - "Populus" should be in italic;
- lines 221 and 224 - "et al." missing comma after period.

Experimental design

The new formulated objective is clear tested using current methodology on polyploids detection (according with bibliographic information for the studied species). However, I suggest the inclusion of a citation in line 107 to support the use of indirect identification of polyploid in early stages using morphological traits. For example, in Escandón et al. (2006), the authors observed statistical differences between diploids and synthetic tetraploids of B. monnieri in some morphological traits using also in this study.

Validity of the findings

No comment

Additional comments

This study highlights the importance to include polyploid induction methodologies to improve the potential of some medicinal plants. The authors correctly justified the reviewers' comments, increasing the manuscript quality.

Reviewer 2 ·

Basic reporting

The authors have revised the manuscript in accordance with the comments given earlier.

Experimental design

The authors have revised the manuscript in accordance with the comments given earlier.

Validity of the findings

The authors have revised the manuscript in accordance with the comments given earlier.

Additional comments

The authors have revised the manuscript in accordance with the comments given earlier. However, there are some minor changes, as highlighted in red fonts.

Annotated reviews are not available for download in order to protect the identity of reviewers who chose to remain anonymous.

---

## Round 0.3 · accepted · Accept

I am happy to say that the authors have addressed all the comments and the revised manuscript gives a very nice impression. I do think it is ready for publication and I congratulate the authors on their nice paper.